# Levodopa Rescues Retinal Function in the Transgenic A53T Alpha-Synuclein Model of Parkinson’s Disease

**DOI:** 10.3390/biomedicines12010130

**Published:** 2024-01-08

**Authors:** Katie K. N. Tran, Vickie H. Y. Wong, Kirstan A. Vessey, David I. Finkelstein, Bang V. Bui, Christine T. O. Nguyen

**Affiliations:** 1Department of Optometry and Vision Sciences, The University of Melbourne, Parkville, VIC 3010, Australia; katiet1@student.unimelb.edu.au (K.K.N.T.); vickie.wong@unimelb.edu.au (V.H.Y.W.); bvb@unimelb.edu.au (B.V.B.); 2Department of Anatomy and Physiology, The University of Melbourne, Parkville, VIC 3010, Australia; k.vessey@unimelb.edu.au; 3The Florey Institute of Neuroscience and Mental Health, The University of Melbourne, Parkville, VIC 3010, Australia; david.finkelstein@florey.edu.au

**Keywords:** A53T mice, levodopa, Parkinson’s disease, electroretinography, optical coherence tomography, amacrine cells, immunohistochemistry

## Abstract

Background: Loss of substantia nigra dopaminergic cells and alpha-synuclein (α-syn)-rich intraneuronal deposits within the central nervous system are key hallmarks of Parkinson’s disease (PD). Levodopa (L-DOPA) is the current gold-standard treatment for PD. This study aimed to evaluate *in vivo* retinal changes in a transgenic PD model of α-syn overexpression and the effect of acute levodopa (L-DOPA) treatment. Methods: Anaesthetised 6-month-old mice expressing human A53T alpha-synuclein (HOM) and wildtype (WT) control littermates were intraperitoneally given 20 mg/kg L-DOPA (50 mg levodopa, 2.5 mg benserazide) or vehicle saline (*n* = 11–18 per group). *In vivo* retinal function (dark-adapted full-field ERG) and structure (optical coherence tomography, OCT) were recorded before and after drug treatment for 30 min. *Ex vivo* immunohistochemistry (IHC) on flat-mounted retina was conducted to assess tyrosine hydroxylase (TH) positive cell counts (*n* = 7–8 per group). Results: We found that photoreceptor (a-wave) and bipolar cell (b-wave) ERG responses (*p* < 0.01) in A53T HOM mice treated with L-DOPA grew in amplitude more (47 ± 9%) than WT mice (16 ± 9%) treated with L-DOPA, which was similar to the vehicle group (A53T HOM 25 ± 9%; WT 19 ± 7%). While outer retinal thinning (outer nuclear layer, ONL, and outer plexiform layer, OPL) was confirmed in A53T HOM mice (*p* < 0.01), L-DOPA did not have an ameliorative effect on retinal layer thickness. These findings were observed in the absence of changes to the number of TH-positive amacrine cells across experiment groups. Acute L-DOPA treatment transiently improves visual dysfunction caused by abnormal alpha-synuclein accumulation. Conclusions: These findings deepen our understanding of dopamine and alpha-synuclein interactions in the retina and provide a high-throughput preclinical framework, primed for translation, through which novel therapeutic compounds can be objectively screened and assessed for fast-tracking PD drug discovery.

## 1. Introduction

Parkinson’s disease (PD) is the second most common neurodegenerative disorder of the central nervous system (CNS) after Alzheimer’s disease [1,2]. There remains no cure for Parkinson’s disease, with the gold-standard treatment for PD continuing to involve the prescription of dopamine replacement therapies [3,4], most commonly levodopa (L-DOPA) and/or dopamine agonists [5]. First introduced as a PD therapeutic in 1967 [6], L-DOPA has been one of the most successful treatments for PD, resulting in dramatic improvements in quality of life and prolonged life expectancies [7,8]. Nevertheless, the efficacy of long-term L-DOPA treatment can change over time as undesirable effects such as dyskinesia [9], potential comorbid anxiety or depression [10,11], drug resistance [12] and shrinkage of therapeutic windows [13,14,15] gradually manifest. Furthermore, L-DOPA therapy provides patients with only transient relief from motor impairment symptoms, leaving PD’s underlying disease processes unperturbed. As such, there is a strong need to identify new disease-modifying therapeutics for PD that can help slow the progression of the disease. A key contributing factor of successful preclinical drug development are dynamic biomarkers that can be used to assess the efficacy and safety of novel compounds. In this study, we use L-DOPA as a proof-of-principle drug to develop a high-throughput preclinical platform that can be used to screen other PD drug candidates.

Parkinson’s disease is well known for its motor manifestations [16,17]; however, patients can manifest a multitude of non-motor symptoms [18,19,20,21], ranging from olfactory desensitisation [22] to autonomic dysfunction [23] to changes in vision [24] by way of PD’s complex pathogenesis profile. Two key pathophysiological hallmarks of Parkinson’s disease include the progressive death of dopaminergic neurons within the substantia nigra pars compacta (SNpc) and the accumulation of alpha-synuclein (α-syn) Lewy bodies (LBs) [25,26] within the central nervous system. Synergistic interactions between dopamine and α-syn within the context of Parkinson’s disease have long been a topic of investigation [27,28]. Some theorise that as toxic α-syn oligomers are stabilised by dopamine [29], LBs deposit in dopamine-rich areas such as the SNpc and peripheral locations like the retina [30,31], leading to the characteristic dopaminergic cell loss reported in these areas in PD. Thus, the selective vulnerability of dopaminergic neurons in the nigrostriatum may have a direct correlate in the retina in PD [32,33].

Given that the eye embryologically develops from the brain [34], the retina is an accessible place to directly probe and evaluate potential changes in neuronal function and structure related to dopaminergic and α-syn interactions in PD. As the most abundant catecholamine within the CNS, dopamine plays an important role in controlling reward and voluntary movement in the brain [35]. Dopamine is also a major neurotransmitter in the retina [36,37], where it is a major regulator of light adaptation [38], eye growth [39], circadian rhythms [40] and neuronal cell survival [41]. Analogous to dopaminergic neurons in the SNpc, a subclass of inhibitory neurons called amacrine cells locally produce and release dopamine within the retina [31,42]. Several groups have reported a loss of amacrine cells and consequent decreased retinal dopamine levels in PD, which have been linked to compromised contrast sensitivity and colour vision [43]. Pharmacodynamic studies which have assessed the effect of L-DOPA on visual outcomes in Parkinson’s patients observe a recovery of retinal function [44,45,46] in addition to improvements in contrast [47] and colour [48] discrimination. As such, retinal biomarkers hold strong potential as dynamic tools for assessing the efficacy and safety [49] of alternative drug candidates for Parkinson’s treatment.

In this study, we investigated the acute effects of L-DOPA treatment on *in vivo* retinal function and structure in a transgenic A53T α-syn mouse model of Parkinson’s disease as a preclinical correlate to clinical findings in the existing literature [45,50]. One of three synuclein alpha (SNCA) missense mutations (A53T, A30P and E46K), the A53T (Ala-53 → Thr) gene has been well established to generate an overexpression of LB-like α-syn deposits within the CNS of homozygous mice [51]. Multiple behavioural studies have established clear age-dependent motor, memory and sensory deficits in A53T mice which mirror impairments in movement and cognition seen at various stages of PD [52,53,54]. Importantly, these behavioural changes have been shown to occur in conjunction with the depletion of tyrosine hydroxylase and dopaminergic cell loss in the SNpc [52,55]. The A53T α-syn mouse model of Parkinson’s disease has previously been studied in chronic drug intervention experiments that examined the long-term effects of iron-chelating agents such as clioquinol [56] and PBT434 [57]. The effect of chronic L-DOPA treatment on cortical iron levels and dopaminergic neurodegeneration in A53T mice has been well characterised, with researchers reporting long-term L-DOPA administration provides a level of neuroprotection by preventing age-related iron accumulation in the SNpc [58]. That said, no study has yet evaluated the acute effects of L-DOPA treatment on retinal function and structure in the A53T murine model of PD. As such, this study quantified *in vivo* retinal function (electroretinography, ERG) and structure (imaging using optical coherence tomography, OCT) in 6-month-old A53T homozygous mice and wildtype littermate controls before and after L-DOPA or saline vehicle treatment. As previously characterised [59], 6-month-old A53T mice manifest moderate outer retinal dysfunction and degeneration correlated with elevated levels of α-syn. Thereby, we aimed to assess the capacity of simple, non-invasive retinal biomarkers such as ERG and OCT to detect and monitor the efficacy of L-DOPA as a proof-of-principle of their utility as preclinical high-throughput screening tools for novel PD therapeutic compounds.

## 2. Materials and Methods

### 2.1. Mice and Levodopa Treatment

A53T mice from the A53T transgenic M83 line (JAX stock #004479; B6; C3-Tg (Prnp-SNCA*A53T) 83Vle/J) were bred at the Melbourne Brain Centre (Kenneth Myer Building, Parkville, VIC, Australia) and housed in adherence to the National Health and Medical Research Council Australian Code of Practice for the care and use of animals for scientific purposes. Every mouse in this experiment was genotyped with two probes to confirm hA53T homozygosity (SNCA-2 Tg and Chr12-3 WT, Transnetyx, Cordova, TN, USA). Additionally, as the retinal degeneration allele Pde6b^rd1^ can be expressed by the background strain of these mice (B6C3H), the Pde6b^rd1^ gene was bred out of our colony and genotyping (Transnetyx) was used to confirm this. All experimental procedures involving animals were approved by The Florey Institute of Neuroscience and Mental Health Animal Ethics Committee (Approval number: 20-043-UM) and performed in accordance with the ARVO Statement for the Use of Animals in Ophthalmic and Vision Research and ARRIVE (Animal Research: Reporting of In vivo Experiments) guidelines [60].

A total of 61 adult (6-month-old, gender-balanced ratio~1:1) A53T homozygous (HOM, *n* = 34) and wildtype (WT, *n* = 27) littermate control mice were included in this study. All animals were fed standard lab chow ad libitum and housed at room temperature (21 °C) on a 12–12 h light–dark cycle. This 6-month age group was chosen to be examined as this stage has been found to manifest retinal dysfunction and degeneration with elevated levels of α-syn [59].

Mice underwent *in vivo* retinal assessment of structure and function (conducted one week apart). Mice were anesthetised intraperitoneally (IP) with a mixture of ketamine (80 mg/kg) and xylazine (10 mg/kg, Troy Laboratory, Smithfield, NSW, Australia) diluted in sterile saline (1:10). Shortly after baseline measurements (5 min), animals were acutely dosed with a 20 mg/kg intraperitoneal injection of either the dopamine precursor 1 ± 3,4dihydroxyphenylalanine (L-DOPA, Madopar^®^, 50 mg levodopa: 12.5 mg benserazide; Roche, NSW, Australia), shown to be effective in rodents at this dose [61], or sterile saline vehicle (VEH, 0.9% sodium chloride, Baxter Healthcare Pty Ltd., Old Toongabbie, NSW, Australia). Proxymetacaine (0.5%) and tropicamide eyedrops (1%, Alcaine™ and Mydriacyl™, respectively, Alcon Laboratories, Frenchs Forest, NSW, Australia) were applied to produce topical anaesthesia and pupil dilation. Ocular surface hydration was maintained through lubricating eye drops or eye gel (Systane^®^ or GenTeal^®^, respectively, Novartis, Macquarie Park, NSW, Australia) and body temperature maintained with a heat pad (37.5 ± 0.5 °C).

### 2.2. Optical Coherence Tomography

*In vivo* retinal structure was measured and quantified by spectral domain optical coherence tomography (SD-OCT, Spectralis^®^, Heidelberg Engineering, Heidelberg, Germany) across all experiment groups (A53T HOM + L-DOPA, *n* = 18; A53T HOM + VEH, *n* = 16; WT + L-DOPA, *n* = 14; WT + VEH, *n* = 13). Retinal volumetric scans (8.1 × 8.1 × 1.9 mm) centred on the optic nerve head were taken at baseline and 30 min after (in follow-up mode) acute L-DOPA or VEH treatment. Each OCT volume scan (121 B-scans, each made up of 768 A-scans; lateral resolution: 3.87 μm axial × 9.8 μm) was collected at a mean speed of 85,000 scans per second with six-frame automated real-time averaging.

Retinal layer thickness profiles (retinal nerve fibre layer, RNFL; ganglion cell inner plexiform layer, GCIPL; inner nuclear layer, INL; outer plexiform layer, OPL; outer nuclear layer, ONL; total retinal thickness, TRT) were automatically segmented using the Heidelberg Eye Explorer 2 OCT reader plugin (Heyex, Heidelberg Engineering). After the correction of any segmentation errors, an annular grid (Early Treatment Diabetic Retinopathy Study [ETDRS] outer 6 mm diameter ring) was placed on the optic nerve head and mean thickness values from the outer ring quadrants of the grid was used for OCT analysis as previously described [59,62,63].

### 2.3. Electroretinography

One week after OCT imaging was conducted, *in vivo* retinal function was assessed in a subset of the same mice using dark-adapted full-field electroretinography (ERG) as previously detailed [63] (A53T HOM + L-DOPA, *n* = 11; A53T HOM + VEH, *n* = 13; WT + L-DOPA, *n* = 11; WT + VEH, *n* = 11). In brief, to maximise retinal sensitivity mice were dark-adapted (>12 h) [64] the night before ERG experiments and a dim red headlight was used during animal preparation. Following anaesthesia and mydriasis, animals were gently secured to a heated platform inside a Faraday cage. Custom-made silver-chloride electrodes (A&E Metal Merchants, Sydney, NSW, Australia) attached to platinum leads (F-E-30, Grass Telefactor, West Warwick, RI, USA) were used for the active/inactive electrodes and a stainless-steel electrode (F-E-30, Grass Telefactor) was employed as the ground electrode as previously reported [59,63].

The effect of L-DOPA treatment over 30 min was determined by repeated recording of the ERG using full-field light stimuli of 2.07 log cd.s/m^2^ at 5 min intervals over a period of 30 min and was achieved through a Ganzfeld bowl (Photometric Solutions International, Oakleigh, VIC, Australia). Baseline ERG recordings were taken from both eyes 5 min before L-DOPA or saline vehicle was delivered. Signals were sampled (Scope™, Powerlab ADInstruments, Bella Vista, NSW, Australia) over 640 ms at 4 kHz and band-pass filtered (0.3 to 1000 Hz, −3 dB).

As previously described, post hoc analysis of the ERG waveforms was carried out in Excel™ (Microsoft, Redmond, WA, USA). Photoreceptoral function (a-wave) was computed by modelling the raw ERG response with a delayed Gaussian function [65] (P3) which returns maximal photoreceptoral amplitude (RmP3, μV). Subtraction of the P3 from the raw waveform gives rise to the P2-OP complex. The P2-OP complex undergoes a discrete Fourier transform and is then digitally filtered to isolate the ON-bipolar cell-driven P2 (b-wave, low pass filter, 46.9 Hz, −3 dB) [66] from which bipolar cell peak amplitude (μV) can be extracted. In this study, we analysed changes in RmP3 and P2 peak amplitude over time after acute L-DOPA/VEH treatment. Temporal ERG data were measured during data collection; however, as there were no clear changes over time or with treatment, we elected not to present ERG timings.

### 2.4. Immunohistochemistry

After *in vivo* measurements, the animals were perfused with phosphate-buffered saline (PBS, 0.1 M solution) and euthanised (100 mg/kg pentobarbitone sodium, Lethabarb, Virbac (Australia) Pty Ltd., Penrith, NSW, Australia). Eyes were enucleated for immunohistochemical assessment. Retinal tissue from each treatment group (*n* = 7–8) was used for quantitative assessment of tyrosine hydroxylase (TH)-positive amacrine cell numbers. Eyes were fixed in 4% paraformaldehyde (PFA, Sigma-Aldrich, St. Louis, MI, USA) diluted in 0.1 M PBS solution for 1 h at room temperature. Eyes were then washed with PBS solution and the anterior segment (including the lens, iris, cornea) was dissected and removed. Separated posterior eyecups were placed in a series of graded (10%, 20% and 30%) sucrose solutions and snap frozen in liquid nitrogen for cryoprotection. Retinae were isolated from the sclera and rinsed in PBS solution before they were incubated with a polyclonal rabbit antibody specific for the detection of tyrosine hydroxylase (1:1000, #AB152, Merck, Kenilworth, NJ, USA) for 3 nights at room temperature. Retinae were then washed and incubated with a secondary goat anti rabbit IgG Alexa 647 antibody (1:500; Thermo Fisher Scientific, Scoresby, VIC, Australia, Cat no. A-21244) at room temperature for 1 h. Retinae were then washed once more, flat-mounted and cover slipped with the photoreceptor side down. Retinae were imaged on the Zeiss LSM800 Airyscan confocal microscope (Carl Zeiss, Jena, Thuringia, Germany) with Zen Blue acquisition software (version 2.3, Carl Zeiss) at 10× magnification. A stitched tile scan was taken of each retina was acquired, with the focal plane adjusted regionally to ensure TH-positive nuclei were in the plane of acquisition. TH-positive amacrine cells were manually counted by researchers (masked to the experimental groups) using FIJI software (National Institutes of Health, Bethesda, MD, USA).

### 2.5. Data and Statistical Analysis

To quantify the effect of L-DOPA on retinal measures in A53T animals over time, statistical comparisons across experiment groups were performed using Prism 8 software (GraphPad Software Inc., San Diego, CA, USA). All data are expressed as group mean ± standard error of the mean (SEM) unless stated otherwise. Trends over time were also analysed by normalisation to baseline values. A ROUT test was used to identify any outliers which were removed from analysis. In most cases, a repeated two-way measures analysis of variance (ANOVA) was used to compare groups with Sidak’s correction for post hoc multiple comparisons at specific times. An alpha of less than 0.05 was deemed statistically significant.

## 3. Results

### 3.1. Acute Levodopa Treatment on Retinal Amacrine Cell Expression in A53T Mice

Whole retina from WT and A53T mice were labelled using immunohistochemistry (IHC) for dopaminergic amacrine cells using an antibody against tyrosine hydroxylase (TH; Figure 1). TH-positive amacrine cell bodies were found in the inner retina and numbers were similar between WT control (Figure 1A,B) and A53T HOM mice (Figure 1C,D), irrespective of acute L-DOPA or vehicle (VEH) treatment (two-way ANOVA; genotype effect, *p* = 0.569; treatment effect, *p* = 0.101; Figure 1E). This suggests that retinal amacrine cell numbers are preserved at the 6-month timepoint in the A53T transgenic mouse model of PD.

### 3.2. Acute Levodopa Treatment on In Vivo Retinal Function in A53T Mice

Electroretinography (ERG) was used to assess any functional effects of short-term L-DOPA treatment (Figure 2 and Figure 3). Group-averaged waveforms of the dark-adapted ERG response to a single 2.07 log cd/m^2^ full-field flash are presented for WT (Figure 2A,C) and HOM mice (Figure 2B,D) treated with vehicle or L-DOPA at baseline (dotted traces) and at 30 min post treatment (solid traces). HOM animals had attenuated photoreceptor (a-wave amplitude, post hoc mean group comparison, genotype effect, *p* = 0.0015, Figure 2E) and inner retinal cell (b-wave amplitude, post hoc mean group comparison, genotype effect, *p* < 0.0001, Figure 2F) responses at baseline compared to WT controls. Acute L-DOPA administration to the A53T HOM group significantly improved photoreceptoral a-wave (RmP3, two-way ANOVA, time × L-DOPA effect, 38–54% increase, *p* = 0.0002, Figure 2E and Figure 3A) and bipolar cell b-wave (P2 peak, two-way ANOVA, time × L-DOPA effect, 23–49% increase, *p* < 0.0001, Figure 2F and Figure 3C) function over time when compared against all other experiment groups. Importantly, post hoc analysis revealed photoreceptoral (*p* = 0.0209, Figure 3B) and bipolar cell (*p* = 0.0106, Figure 3D) responses in HOM animals treated with L-DOPA grew more over time compared to HOM mice treated with vehicle. ERG responses in WT and HOM animals before and after VEH or in WT animals treated with L-DOPA largely remained the same if not attenuated after 30 min (Figure 2A–C).

### 3.3. Acute Levodopa Treatment on In Vivo Retinal Structure in A53T Mice

Optical coherence tomography (OCT) imaging was used to compare *in vivo* retinal structure before and after acute L-DOPA or saline vehicle (VEH) treatment in A53T HOM and WT mice (Figure 4). Consistent with previous observations [59], the inner retinal layers were not affected in HOM animals (Figure 4C–H); however, HOM mice had thinner outer retinal layers than WT controls, namely at the outer plexiform (OPL, post hoc mean group comparison, genotype effect, *p* = 0.0009, Figure 4I,J) and outer nuclear layers (ONLs, post hoc mean group comparison, genotype effect, *p* < 0.0001, Figure 4K,L).

We observed global non-specific thinning of retinal layers across time (30 min) in most experiment groups, including in the RNFL (both WT and HOM animals, two-way ANOVA, time effect, *p* < 0.0001, Figure 4C,D), GCIPL (both WT and HOM animals, two-way ANOVA, time effect, *p* < 0.0001, Figure 4E,F), INL (HOM animals only, two-way ANOVA, time effect, *p* = 0.0009, Figure 4H), OPL (HOM animals only, two-way ANOVA, time effect, *p* < 0.0001, Figure 4J), ONL (both WT and HOM animals, two-way ANOVA, time effect, *p* = 0.007 to 0.0137, Figure 4K,L) and thickness of the whole retina, TRT (both WT and HOM animals, two-way ANOVA, time effect, *p* < 0.0001, Figure 4M,N). Interaction effects in GCIPL and OPL thickness (two-way ANOVA, interaction effect, *p* = 0.0107 to 0.0384, Figure 4E and Figure 4I, respectively) were found in WT animals. This suggests changes in retinal thickness occur over time under general anaesthesia.

To overcome this global effect on retinal thickness of imaging over time, comparison of L-DOPA/VEH effect between genotypes was made following normalisation to baseline OCT (Figure 5). Here, we found that changes in retinal layer thickness over time were similar across all experiment groups with the exception of the outer plexiform layer (OPL), where a genotype effect was found, with the OPL thinning more in HOM animals than their WT counterparts (two-way ANOVA, genotype effect, *p* = 0.0041, Figure 5D). Importantly, OCT image quality did not change across time or with acute L-DOPA treatment (Appendix A).

## 4. Discussion

Our study is the first to evaluate the effect of acute L-DOPA treatment on *in vivo* retinal function and structure in the A53T mouse model of PD. We show that at 6 months of age, TH immunopositive amacrine cell numbers remain intact in A53T animals across genotype despite HOM mice exhibiting reduced retinal function and outer retinal thinning. Additionally, we show acute L-DOPA treatment improves retinal function in A53T HOM animals, enhancing our understanding of the interactions between dopamine and alpha-synuclein in the retina. Together, these findings aid in establishing a non-invasive preclinical framework that facilitates the translation of innovative therapeutic compounds, enabling their objective and rapid screening and evaluation for PD.

### 4.1. Amacrine Cells Are Preserved in 6-Month-Old A53T Mice

Dopaminergic, or A18, amacrine cells, with their cell bodies localised mainly to the inner plexiform layer, stained strongly for TH [42,67] and were found to be spread evenly across the retina of both WT and HOM mice at 6 months of age. Whilst TH-positive neuronal loss has been reported in the substantia nigra pars compacta (SNpc) of A53T mice older than 6 months of age [52,55,68] (8–12 months), we did not observe a difference in TH-positive amacrine cell numbers in 6-month-old A53T mice (Figure 1E). In the retina, a reduction in dopaminergic amacrine cell number shortly after the induction of an intravitreal adeno-associated viral injection model of alpha-synuclein overexpression has been observed previously [69]. Thus, it is likely that TH-positive amacrine cells would be lost in HOM retina at later timepoints. In line with this, we have previously shown that 6-month old A53T HOM mice have elevated tyrosine hydroxylase (TH) levels compared to WT controls [59], suggesting there are abnormalities in dopaminergic signalling in A53T mice at 6 months of age that occur in the absence of dopaminergic amacrine cell loss.

### 4.2. Acute L-DOPA Treatment Ameliorates Retinal Dysfunction in A53T Mice

In terms of retinal function, we demonstrate for the first time using full-field electroretinography (ERG) that acute L-DOPA treatment improves retinal function in A53T HOM mice. Mirroring reports of attenuated flash full-field ERG responses in clinical PD cohorts [46,70,71,72] and our previous preclinical observations [59], we show that A53T HOM animals have significantly smaller photoreceptoral and bipolar cell responses than their WT counterparts (Figure 2E,F). While acute L-DOPA treatment does not change ERG responses in WT animals (Figure 2C), the a-wave (Figure 2E and Figure 3A) and b-wave (Figure 2F and Figure 3B) of L-DOPA-treated HOM animals grew approximately 20% more than vehicle (VEH) saline treated HOM animals. These findings are in broad agreement with Marocco et al. [69] who observed an improvement in light-adapted ERG responses in their intravitreal adeno-associated viral mouse model of α-syn overexpression after systemic injections of L-DOPA. Our findings recapitulate those of retinal pharmacodynamic L-DOPA studies within the PD clinical literature which report an amelioration of photoreceptor and bipolar cell ERG responses following acute L-DOPA administration under complete wash-out conditions and/or in PD patients naïve to dopamine replacement treatment [45,46,71]. Given the characteristic loss of dopaminergic amacrine cells in PD patients [73], a transient influx of systemic dopamine is likely to improves retinal function by way of increasing dopaminergic neurotransmission at D_1_ and D_2_ receptors [38,74]. Considering there is evidence to suggest that subchronic L-DOPA administration can reduce the accumulation of central phosphorylated alpha-synuclein in the SNpc [75], future retinal studies quantifying and/or correlating electrophysiological and immunohistochemical outcomes in A53T mice following chronic L-DOPA treatment would contribute to strengthening our understanding of dopamine and alpha-synuclein’s synergistic interactions within the context of Parkinson’s disease.

### 4.3. Retinal Thinning Occurs in A53T Mice after Acute L-DOPA Treatment

To our knowledge, this is the first time retinal structure has been monitored using OCT imaging before and after acute L-DOPA treatment in the A53T mouse model of PD. Akin to reports of photoreceptor layer thinning in the clinical human PD literature [76,77,78,79,80], we show that A53T HOM animals have thinner outer plexiform (OPL, Figure 4I,J) and outer nuclear (ONL, Figure 4K,L) layers compared to WT controls. These findings are also in agreement with a previous study by our group [59]. In general, we find that acute L-DOPA treatment largely does not change retinal thickness; however, we discovered that most retinal layers indiscriminately thinned over time with both VEH and L-DOPA intervention. Given that OCT image quality remained stable across the experimentation window (Appendix A), we postulate that these changes in tissue thickness may be due to water loss caused by prolonged anaesthesia-induced dehydration [81].

## 5. Conclusions

In summary, this study is the first to investigate and track the acute effects of L-DOPA treatment on retinal function and structure in A53T transgenic mice. While IHC revealed no change in amacrine cell number in A53T groups at 6 months, we demonstrate that retinal dysfunction and outer retinal thinning occurs in this mouse model of α-syn overexpression. These findings highlight the potential for *in vivo* ocular phenotyping of A53T mice to assess Parkinson’s disease changes prior to overt loss of TH-amacrine cells. Additionally, we show L-DOPA dynamically improves attenuated scotopic photoreceptor and bipolar cell ERG responses in A53T mice while not affecting WT responses. Our findings indicate that future clinical Parkinson’s disease biomarker studies should consider L-DOPA status when conducting ERG examinations, whereas OCT may be more immune to acute L-DOPA treatment. Overall, our findings deepen our understanding of the interaction between dopamine, alpha-synuclein and the retina, supporting the utility of the ERG in A53T mice as a high-throughput preclinical platform through which new therapeutic compounds can be objectively screened and assessed.

## Figures and Tables

**Figure 1 biomedicines-12-00130-f001:**
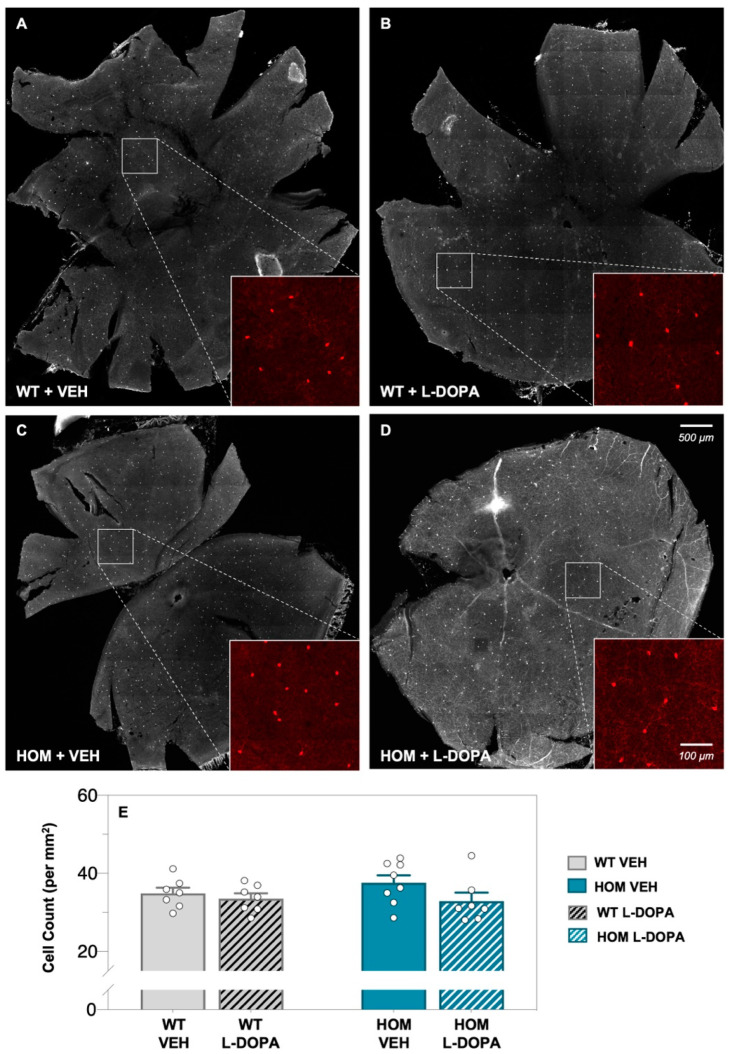
**Effect of L-DOPA on dopaminergic amacrine cells in A53T retinae.** Representative retinal flatmount images (×10 magnification) of dopaminergic tyrosine hydroxylase (TH)-positive amacrine cells of vehicle (VEH)- and levodopa (L-DOPA)-treated wildtype (WT, (**A**,**B**)) and A53T homozygous (HOM, (**C**,**D**)) cohorts. Scale bars: outside inset = 500 μm, inside inset = 100 μm. Group average (±SEM) TH-positive amacrine cell counts (**E**) of all experiment groups. Gray bars denote WT animals (plain, VEH treated, *n* = 7; black-striped, L-DOPA treated, *n* = 7) and teal bars denote A53T HOM mice (plain, VEH-treated, *n* = 8; white-striped, L-DOPA-treated, *n* = 7). All data shown are mean ± SEM; repeated two-way ANOVA analyses found no difference in amacrine cell numbers across all four experiment groups.

**Figure 2 biomedicines-12-00130-f002:**
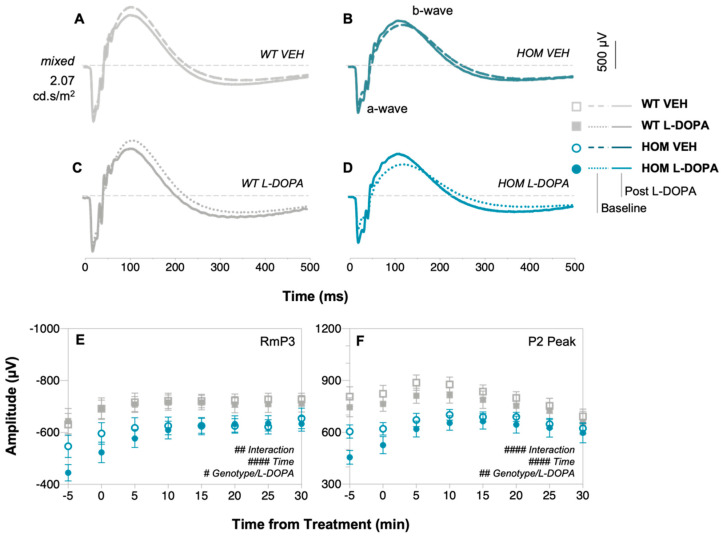
**In vivo retinal function of A53T animals following acute L-DOPA treatment.** Group-averaged scotopic electroretinogram (ERG) waveforms of (**A**) wildtype (WT) mice treated with vehicle (VEH) saline solution, (**B**) A53T homozygous (HOM) mice treat with VEH, (**C**) WT mice treated with L-DOPA and (**D**) HOM mice treated with L-DOPA. Dotted traces represent ERG responses at baseline, while solid traces denote ERG responses 30 min after treatment administration. Raw RmP3 (**E**) and rod P2 (**F**) amplitudes, tracked at 5 min intervals across 30 min. Gray traces/symbols denote WT animals (light/unfilled, VEH treated, *n* = 11; dark/filled, L-DOPA treated, *n* = 11) and teal traces/symbols denote A53T HOM mice (dark/unfilled, VEH-treated, *n* = 13; light/filled L-DOPA-treated, *n* = 11). All data shown are mean ± SEM; # *p* < 0.05, ## *p* < 0.01, #### *p* < 0.0001: effect(s) on repeated two-way ANOVA analyses.

**Figure 3 biomedicines-12-00130-f003:**
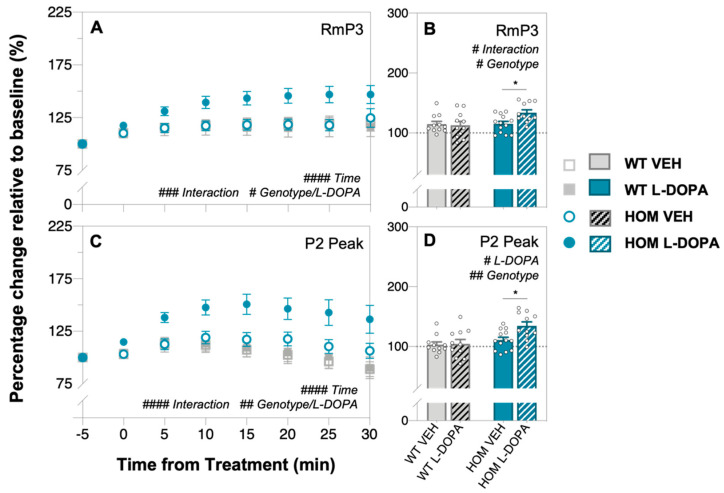
**In vivo retinal function in A53T animals expressed relative to baseline following acute L-DOPA treatment.** Percentage (%) change relative to baseline of ERG responses (RmP3, (**A**,**B**) and P2 peak, (**C**,**D**)) over time. A53T HOM animals treated with L-DOPA exhibited the greatest growth in ERG responses compared to all other experiment groups. Gray bars/symbols denote WT animals (plain, VEH treated, *n* = 11; black-striped, L-DOPA treated, *n* = 11) and teal bars/symbols denote A53T HOM mice (plain, VEH-treated, *n* = 13; white-striped, L-DOPA-treated, *n* = 11). All data shown are mean ± SEM; # *p* < 0.05, ## *p* < 0.01, ### *p* < 0.001, #### *p* < 0.0001: effect(s) on two-way ANOVA analyses; * *p* < 0.05: L-DOPA effect on post hoc analyses; dotted line indicates 100% mark.

**Figure 4 biomedicines-12-00130-f004:**
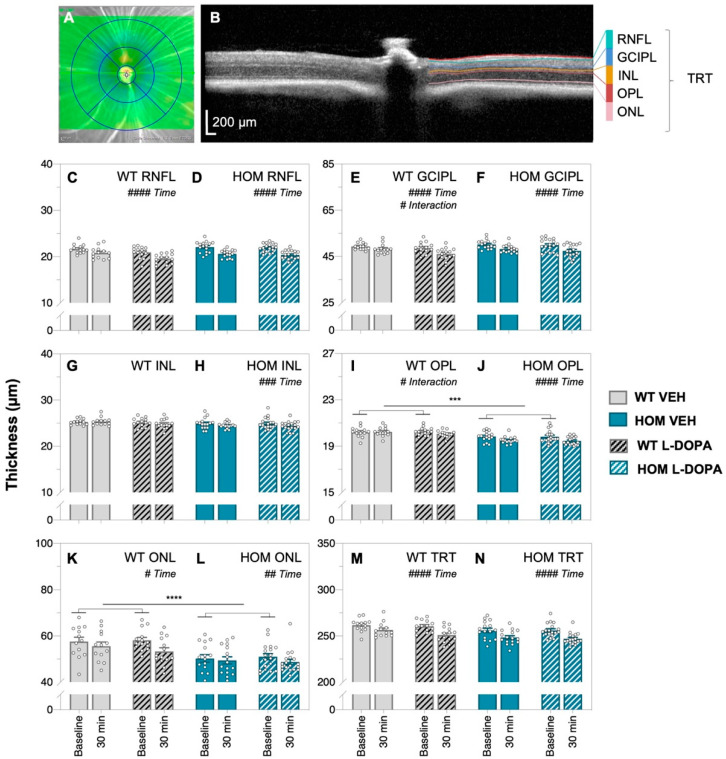
**In vivo retinal structure of A53T animals following acute L-DOPA treatment.** Representative *en face* mouse fundus image (**A**) where the green line indicates the position of the B-scan, and (**B**) a retinal cross-section with the automatic segmentation of retinal layers; scale bar, 200 μm. Comparison of raw retinal thickness values of the retinal nerve fibre layer, RNFL (**C**,**D**); ganglion cell and inner plexiform layer, GCIPL (**E**,**F**); inner nuclear layer, INL (**G**,**H**); outer plexiform layer, OPL (**I**,**J**); outer nuclear layer, ONL (**K**,**L**) and total retinal thickness (TRT, (**M**,**N**)), respectively, of wildtype (WT) and A53T homozygous (HOM) mice at baseline and 30 min after treatment. Generally, the retina in all experiment groups seems to thin across time. A53T HOM animals exhibited thinner OPL and ONL compared to WT counterparts. Gray bars denote wildtype (WT) animals (plain, VEH treated, *n* = 13; black-striped, L-DOPA treated, *n* = 14) and teal bars denote A53T homozygous (HOM) mice (plain, VEH-treated, *n* = 15; white-striped, L-DOPA-treated, *n* = 18). All data shown are mean ± SEM; # *p* < 0.05, ## *p* < 0.01, ### *p* < 0.001, #### *p* < 0.0001: effect(s) on repeated two-way ANOVA analyses; *** *p* < 0.001, **** *p* < 0.0001: genotype effect on post hoc analyses.

**Figure 5 biomedicines-12-00130-f005:**
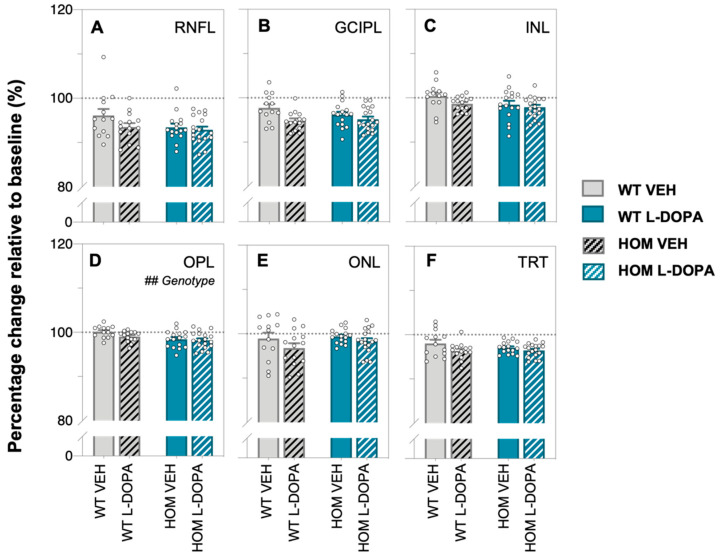
**In vivo retinal structure in A53T animals expressed relative to baseline following acute L-DOPA treatment.** Percentage (%) change relative to baseline of retinal layer thickness values (retinal nerve fibre layer, RNFL, (**A**); ganglion cell and inner plexiform layer, GCIPL, (**B**); inner nuclear layer, INL, (**C**); outer plexiform layer, OPL, (**D**); outer nuclear layer, ONL, (**E**) and total retinal thickness, TRT, (**F**)), respectively, over time. Gray bars denote WT animals (plain, VEH treated, *n* = 13; black-striped, L-DOPA treated, *n* = 14) and teal bars denote A53T HOM animals (plain, VEH-treated, *n* = 15; white-striped, L-DOPA-treated, *n* = 18). All data shown are mean ± SEM; ## *p* < 0.01: effect(s) on two-way ANOVA analyses; dotted line indicates 100% mark.

## Data Availability

The raw data supporting the conclusions of this article will be made available by the authors, without undue reservation.

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
