# Peer review of "Levodopa Rescues Retinal Function in the Transgenic A53T Alpha-Synuclein Model of Parkinson’s Disease"

_biomedicines, 2024, doi:10.3390/biomedicines12010130_

Round 1
Reviewer 1 Report
Comments and Suggestions for Authors
The manuscript “ Levodopa Rescues Retinal Function in the Transgenic A53T Al- 2 pha-synuclein Model of Parkinson’s Disease” , aimed to evaluate in 15 vivo retinal changes in a transgenic PD model of α-syn overexpression and the effect of acute levo- 16 dopa (L-DOPA) treatment. The manuscript is well written and well designed. However , a major concern is that I can’t catch the method through which you induced Parkinson’s disease in mice and what are the successive signs for this induction
Some minor points are required to improve the status of the manuscript.
- Abstract : the experimental design is not well written and needs detailed revision.
- Lines from 46-48: have no citations while it introduces very important information.
- retinal assessment of structure and function “ you need to add citations for the methods , and the detected measures in the methods , specify how did you selected these measures and specify each one according to the related function in the discussion.
- The rationale for the selected dose of Levodopa should be illustrated with references.
- No references were added for the immunohistochemical staining and the type and sources of the primary and secondary antibodies should be mentioned.
- In the figure legends , different “n”, why?
- P value ??? is not presented in the figure legend.
- The limitations of the study should be mentioned.
Comments on the Quality of English LanguageModerate Editing is required.
Reviewer 2 Report
Comments and Suggestions for Authors
Comments on the manuscript “Levodopa Rescues Retinal Function in the Transgenic A53T Alpha-synuclein Model of Parkinson’s Disease” submitted to ‘Biomedicines MDPI’
Comments
Overall, the manuscript is well written and within the merit for publication. However, I have the following major concerns.
§ The starting sentences of the 2nd and 3rd paragraphs of introduction should be rephrased. Starting paragraph with “Apart from its….” Or “As an embryological outpouching…” like sentences is not looking good in academic writing.
§ Writing “in vivo” should be “in-vivo”
§ In animal section, the authors should explain the guidelines, like how the animals were euthanized post-experiment. In 61 mice, what was the ratio of male/female.
§ In Figure 1, the graph below showing the Cell Count showed be labelled clearly “E”
With these few minor points, I accept this manuscript for publication.

Minor correction
Reviewer 3 Report
Comments and Suggestions for Authors
An outstanding paper. Congratulations for the group.
The paper presents a groundbreaking exploration into the potential of levodopa in mitigating the effects of Parkinson's disease on retinal function.
The study into a crucial aspect of Parkinson's disease pathology that has been relatively understudied—its impact on retinal health.
The significance of this paper lies not only in its meticulous methodology, but also in its implications for Parkinson's disease diagnosis and therapy.
Through a series of exceedingly complicated experiments and analyses, they discerned the restorative effects of levodopa on retinal function in this model.
Moreover, the paper's rigor in experimental design, data collection, and statistical analysis reinforces credibility and reliability of its findings. The researchers have taken meticulous care in their approach, and the conclusions drawn are well-supported and scientifically sound.
Hence, I wholeheartedly recommend the publication of this paper for its academic rigor, innovative findings, and its translational potential.
Round 2
Reviewer 1 Report
Comments and Suggestions for Authors
The authors addressed the required reviews and the manuscript is now improved.
Comments on the Quality of English LanguageMinor English Editing is required.